# A predictive approach to integrating connectivity into landscape scale protected areas planning

**Paul O'Brien**⊙*, **Natasha Carr, Jeff Bowman**

Ontario Ministry of Natural Resources (MNR), Peterborough, Ontario, Canada

* obrienp1@myumanitoba.ca

## Abstract

It has been recognized that protecting at least 30% of terrestrial lands by 2030 (30x30) in systems of well-connected protected areas will be necessary for halting global biodiversity loss. While significant effort has been made to increase the coverage of global protected areas, connectivity has been found to fall short of what is needed to maintain proper ecological function. We present methods to explicitly incorporate connectivity into planning of future protected areas, using the protected area network in Ontario, Canada as a case study. We first carried out a series of connectivity analyses with simulated, generic parks, and estimated the effect of each new park on network connectivity. We then built and evaluated a set of regression models to determine characteristics of these generic parks that are predictors of optimal protected area placement for improving network connectivity. We found that in all cases, adding a generic park resulted in improved network connectivity; however, some park placements lead to greater improvements. The top model suggested that protected areas should have the greatest benefit to connectivity of the network when they have a low edge-to-area ratio, are close to the center of the landscape and to a network node, have a low degree of internal anthropogenic influence, but a close proximity to developed landscapes. We demonstrated an application of this approach by using the model to prioritize an independent set of candidate parks by their predicted benefit to network connectivity. The resulting model provides a tool with which Indigenous communities, land trust managers, and protected area planners can evaluate and prioritize candidate sites based on their expected benefit to improving connectivity of a given protected area network. As efforts to increase area-based conservation are ramped up to help meet national 30x30 targets, this should be an important tool for ensuring connectivity is explicitly incorporated into protected areas planning.

## Introduction

Globally, biodiversity faces a multitude of interconnected threats, with habitat loss and land use changes continuing to be among the top drivers of loss [1,2]. Experts

**Data availability statement:** Input data and annotated code used to build the predictive model for candidate site evaluation can be found at Figshare: 10.6084/m9.figshare.27307335.

**Funding:** Funding for this research was provided by the Ontario Ministry of the Environment, Conservation, and Parks and the Ontario Ministry of Natural Resources. The funders had no role in study design, data collection and analysis, decision to publish, or preparation of the manuscript.

**Competing interests:** The authors have declared that no competing interests exist.

estimate that continued habitat loss would threaten or drive up to 80% of species to extinction [2]. Protected areas have long been considered a cornerstone of conservation and it is well agreed upon that significant expansion of the world protected area estate is needed to halt and reverse major biodiversity loss. While effectively placed protected areas can benefit biodiversity by preventing habitat loss, protected areas on their own are often not enough. Wildlife faces increasing pressure from anthropogenic development both within [3] and outside park boundaries [4]. Increased isolation of protected areas due to land-use changes in the surrounding landscape can increase the risk of populations within protected areas becoming isolated [5]. Thus, maintaining and enhancing connectivity among protected areas is essential for the long-term persistence of biodiversity.

It has been internationally recognised across multiple global agreements that ecological connectivity is critical for halting and reversing biodiversity loss (Convention on Biological Diversity), allowing for species migrations (Convention on Migratory Species), and for resilience and adaptation in the face of climate change (United Nations Framework Convention on Climate Change). Goal A of the recent Kunming-Montreal Global Biodiversity Framework (K-M GBF) explicitly mentions connectivity stating, "The integrity, **connectivity**, and resilience of all ecosystems are maintained, enhanced, or restored substantially increasing the area of natural ecosystems by 2050." [6]. Further, Target 3 of the K-M GBF calls for parties to effectively conserve 30% of lands and waters in ecologically representative, **well-connected**, and equitably managed systems of protected areas and other effective conservation measures (OECMs). Creating networks of protected areas linked by ecological corridors can prevent habitat loss, allow for unimpeded movement of animals and flow of ecological processes [7], and allow species to track suitable climates in the face of climate change [8,9].

Indeed, the importance of connectivity is evident from the vast body of research that exists and the wide range of methods for measuring connectivity [10]. Despite being well researched however, connectivity is rarely incorporated into protected areas planning [11]. In a global assessment of protected areas connectivity, Saura et al. [12] found only 7.5% of global lands were protected and connected, while 14.7% of land area was protected. Similarly, Ward et al. [13] found that less than 10% of global protected areas were structurally connected. This could in part be a result of protected areas coverage being historically biased towards remote areas and regions not in conflict with resource extraction [14–16]. While placement of protected areas in remote places does not necessarily mean low connectivity (in fact, likely the opposite given the intactness of many remote places), significant human modification outside of parks and lack of coordination between multiple planning jurisdictions has led to low connectivity across the global protected area estate [13]. It has also been suggested that the absence of connectivity in planning may also stem from ambiguous language used in various international targets (e.g., "well-connected") and a lack of guidance for practitioners and decision-makers on how to measure protected areas connectivity [12,17]. Brodie et al. [18] suggest that a protected area network is "well-connected" if it allows sufficient movement to ensure long-term persistence of

focal taxa, ecosystem function, or ecosystem services when compared to landscapes without barriers. Here, we develop methods to evaluate characteristics of protected areas that benefit the connectivity of protected areas networks, with a goal of providing guidance on measuring protected areas connectivity and incorporating connectivity into planning.

A large variety of methods have been developed for measuring connectivity, and in particular connectivity of protected areas. These methods often fall along a gradient between measuring structural connectivity [13,19,20] and functional connectivity [21–23]. We used the recently developed sentinel node method for evaluating protected areas connectivity and the Mean Pairwise Effective Resistance (MPER) connectivity indicator [24]. This method makes use of circuit theory to model connectivity [25] and differs from other methods modelling protected areas connectivity (e.g., [22,26]) by the introduction of sentinel nodes. Sentinel nodes are a fixed set of nodes placed within a subset of protected areas in the network that are used to evaluate connectivity for a given protected area network. The fixed nature of the sentinel nodes allows estimates of connectivity to be comparable across space and time, despite changes made to the network. The MPER indicator is calculated between the sentinel nodes and measures the average cost (e.g., for an animal) of moving throughout the network. The lower the MPER value, the lower the cost of movement, and thus greater the connectivity across the network. The sentinel node method can be used to measure and track protected areas connectivity through time.

Critical to creating "well-connected" systems of protected areas is being able to integrate connectivity into future planning, which requires systematic prioritization of potential sites to maximize effectiveness of new protected areas [12,27]. We address this need by expanding on the work of O'Brien et al. [24] using the sentinel node method and MPER indicator. Our objective was to produce a method for incorporating connectivity into future protected areas planning and develop a model for practitioners to help prioritise specific locations for protection. Specifically, we sought to ask whether 1) certain characteristics of protected areas influence the connectivity benefit of a given protected area; and 2) we can use these characteristics to help rank candidate protected areas by their potential to improve connectivity of a protected areas network. We illustrate our method using the province of Ontario, Canada, and its protected areas network, as a study area. Our method can serve as a tool for protected areas planners to explicitly incorporate connectivity into the planning process. As efforts ramp up to reach 30 x 30 targets, our model can help prioritize candidate sites to ensure new protected areas contribute to building 'well-connected' systems of protected areas.

## Methods

### Protected area simulations

To understand how connectivity gains may vary with placement of new protected areas depending on their characteristics, we ran a series of connectivity analyses for the addition of different simulated, generic parks compared to a connectivity estimate for a real protected areas network without any added parks. We conducted our analyses following the sentinel node methods outlined by O'Brien et al. [24] for evaluating connectivity of protected area networks. We focused this analysis on the southern region of Ontario, Canada and its protected areas network (Fig 1). We randomly selected 20 protected areas within the study area to serve as sentinel nodes for our analyses to model effects of park placement. Protected areas were considered for node selection so long as they were at least the size of a pixel (i.e., $300m^2$). A random selection procedure was employed to ensure a relatively even distribution of nodes across the study area rather than selecting park nodes based on other criteria (e.g., size), which may lead to biases in node location and a clustering of nodes. We considered 20 sentinel nodes should be sufficient for our study area following recommendations by Koen et al. [28] who found 15–20 nodes to be sufficient for omnidirectional connectivity modelling in a similar region. The sentinel nodes were contained within the boundaries of current protected areas and were used to evaluate connectivity of the region's full protected area network, as represented by the Canadian Protected and Conserved Areas Database (CPCAD; A national database which aggregates spatial data for national, provincial/territorial, and private protected areas as well as OECMs) [29]. Thus, the region contained 730 protected areas and a subset of 20 of these protected areas were sentinel nodes in the full network. We used circuit theoretic models of connectivity [30], which required a cost-to-movement surface,

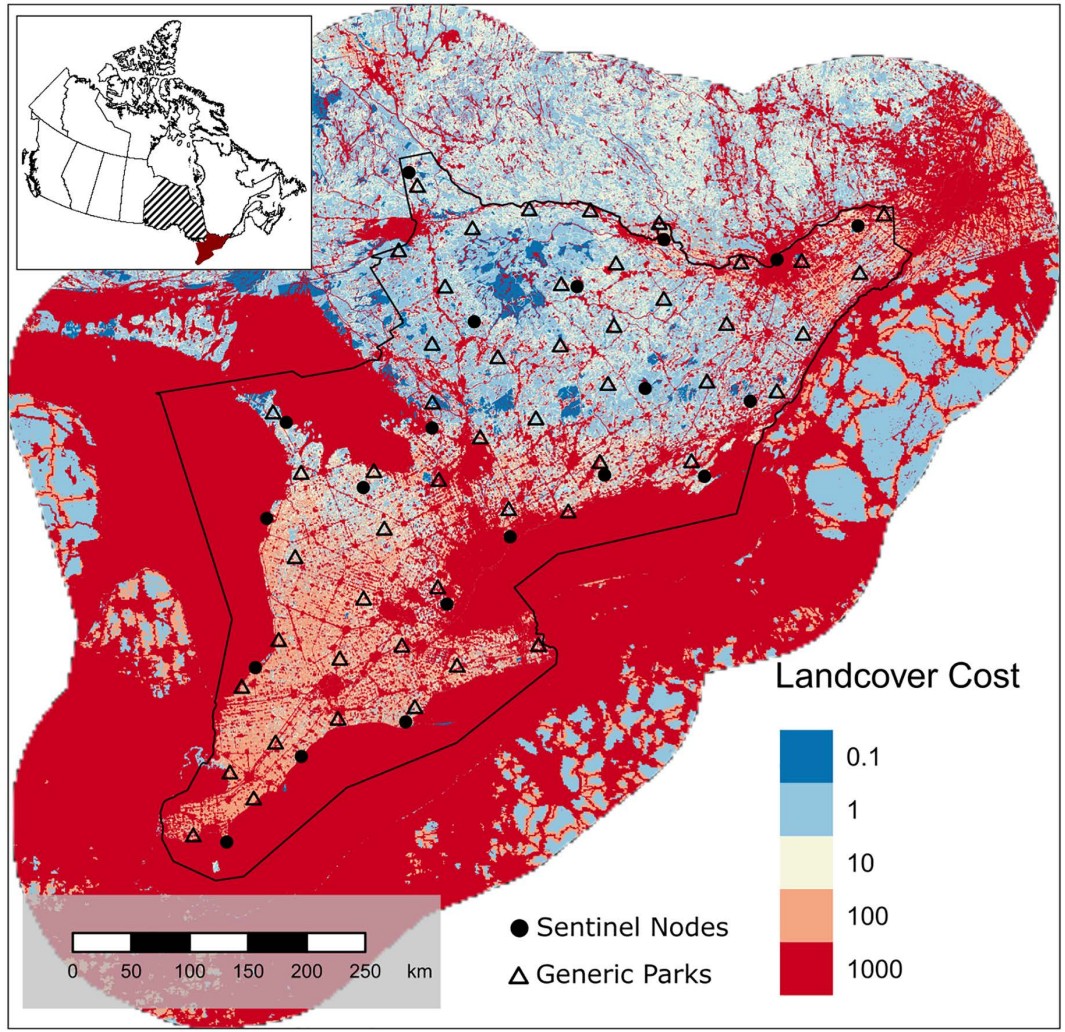

**Fig 1. A cost map for southern Ontario, Canada.** Five cost values were assigned to landscape features based on the degree to which they facilitate or impede movement for terrestrial species that use natural cover. Natural areas within protected areas boundaries were assigned the lowest cost (0.1) under the assumption that they are less costly to move through than natural areas outside of protected areas (S1 Table). The solid black line shows the boundaries of the study area and solid black circles show the locations of the 20 sentinel nodes used to evaluate protected areas connectivity in southern Ontario. Open triangles show the locations of the generic parks used to evaluate optimal park placement for improving connectivity. The modified cost surface was derived from the 300-m resolution cost surface of Canada from Pither et al. [31]. The inset map shows the location of the study area (dark red) within Ontario (hatched). Shapefiles acquired from Ontario Geohub and from Statistics Canada and are licensed under the Open Government License–Ontario and Open Government Licence–Canada, respectively.

representing landscape features by their permeability to animal movement. O'Brien et al. [24] used the 300-m resolution cost surface produced by Pither et al. [31] and validated by Brennan et al. [32]. O'Brien et al. [24] modified this surface by assigning the lowest cost value (0.1) to natural landcover within protected areas, resulting in five cost ranks: 0.1, 1, 10, 100, and 1000 (S1 Table). They made the assumption that natural areas within protected areas are less costly to move through for an animal than natural areas outside protected areas given the regulation of certain human activities within park boundaries (e.g., hunting, resource extraction, etc.). This is an integral aspect of the sentinel node method. Therefore, we used the 300-m cost surface of Ontario modified by O'Brien et al. [24] for the following analyses. The cost surface was clipped to the extent of southern Ontario (Fig 1).

We aimed to evaluate 50 new, generic parks, which we considered should provide a sufficient sample for our modelling procedure and balance computational demands. Therefore, we employed a stratified random procedure to select 50 spatial points within our southern Ontario study area to serve as centroids for generic parks. The algorithm we used applied a stratum to ensure selected points were at least 50 km apart and so the random selection converged when a non-overlapping set of 50 points was reached. Additionally, the 50 random points were drawn from pixels classified as natural (cost = 1) thus, the simulated parks could contain a mix of natural and higher cost pixels, but all had centroid pixels with a cost of 1. The stratified random selection ensured generic parks were both distributed across the study area and that there was no overlap between parks when calculating landscape variables. Boundaries for the generic parks were created by adding buffers to the 50 spatial points. All generic parks were given the same area (area = 6.25 km$^2$ based on the mean size of current protected areas in the study area); however, half ($n = 25$) were assigned a square shape, and the other half assigned a rectangular shape to examine the effect of park shape on connectivity. We visually inspected the set of generic parks by plotting distributions of landscape variables (Table 1) to verify the selection represented a variety of possible protected area additions and thus would provide a range of potential connectivity outcomes. Like the current protected area network (including sentinel node parks), the generic parks were incorporated into the cost-to-movement surface by lowering the cost of natural landcover within park boundaries (from 1 to 0.1), however, they differed from sentinel node parks in that they were not added as nodes in the network. This means that adding a generic park to the landscape, even without making it a node in the network, should improve connectivity of the network as a whole given an overall reduction in cost.

Using the Julia implementation of Circuitscape [25,33], we first evaluated connectivity for the current network of protected areas in southern Ontario (including 20 sentinel nodes and all protected areas from the CPCAD [29]) without the addition of any new sites and we calculated the mean pairwise effective resistance (MPER) as an indicator of connectivity [24]. Pairwise effective resistance, an output of Circuitscape analyses, is a measure of the cost of travelling between two pairs of nodes and is calculated between all pairs of nodes. The MPER indicator is the mean of all the pairwise effective resistance estimates and provides a measure of overall network connectivity [24]. The calculation of MPER for the current protected areas network provided a baseline estimate of connectivity for our southern Ontario study region. Next, we ran a series of connectivity analyses where we independently added each of the 50 simulated, generic parks to the cost surface (by lowering the cost of natural areas within a given parks boundaries) and calculated MPER for each scenario, resulting in 50 scenarios, and 50 unique estimates of MPER for the addition of each generic park to the study area. Lastly, we calculated the ΔMPER, which is the difference between MPER for a given generic park addition and the baseline MPER with no added parks. The ΔMPER measures the change in connectivity of the network from adding a generic park, where positive values of ΔMPER represent a decrease in connectivity and negative values represent an increase in connectivity.

### Predicting optimal protected area placement

To determine what landscape and park characteristics best predict placement of protected areas for improving network connectivity, we evaluated a set of linear regression models with ΔMPER as the response variable. We log-transformed the ΔMPER response variable to help reduce skewness of the data and to meet model assumptions (e.g., normality and homogeneity of residuals). We determined a set of 8 different predictor variables that we posited would influence the effect a given protected area has on improving connectivity (Table 1). This set of predictors included: edge-to-area ratio, distance to centre of the study area, distance to the nearest park, distance to the nearest node, proportion of anthropogenic development surrounding a protected area, proportion of water surrounding a protected area, proportion of anthropogenic development within protected areas, and proportion of water within protected areas. To establish which scale was most appropriate to examine the effects of anthropogenic development and water surrounding parks, we calculated these variables within 4 different buffer sizes around parks: 2 km, 10 km, 25 km, and 50 km (i.e., within the doughnut-shaped area between the park boundary and the outside of the buffer). To evaluate which set of predictors best explained variation

**Table 1. List of predictors used in linear regression models, descriptions of what they measure, and how they were calculated.**

| Variable type | Variable name | Description |
|---|---|---|
| Shape | Edge-to-area ratio (EAR) | The ratio of the edge length (i.e., perimeter (km)) of a generic park to the area (km$^2$) of the park. Provides a measure of park shape.<br>EAR = perimeter of generic park/area of generic park |
| Distance | Distance to centre | The Euclidian distance (km) from the centre of a generic park to the centre of the study area. |
| | Distance to nearest park | The shortest distance (km) between the boundaries of a generic park and the nearest real protected area. |
| | Distance to nearest node | The shortest distance (km) between the boundary of a generic park to the nearest sentinel node. |
| Landscape Context | Proportion of anthropogenic – buffer | The proportion of anthropogenic high cost (cost = 1000) pixels within a buffer surrounding generic parks. Four different buffer sizes were tested (2 km, 10 km, 25 km, and 50 km) to determine the appropriate scale to measure this variable. Calculation only included the buffer area around the park and not the park area itself (i.e., donut area).<br>PropAnthro$_{buffer}$ = # anthro pixels in buffer/total # pixels in buffer |
| | Proportion of water – buffer | Same as above description for anthropogenic high cost, but for high cost due to lakes or rivers (i.e., natural high cost). |
| | Proportion of anthropogenic – within parks | The proportion of anthropogenic high cost (cost = 1000) pixels within the boundaries of a generic park.<br>PropAnthro$_{park}$ = # anthro pixels in park/total # pixels in park |
| | Proportion of water – within parks | Same as above for anthropogenic high cost, but for high cost due to lakes or rivers. |

in ΔMPER, we constructed a set of 10 candidate models using data from the 50 generic protected areas (S2 Table). We used Akaike Information Criterion adjusted for small sample sizes (AIC$_c$) to select the best model of our 10 candidates for predicting park placement that should maximize connectivity gains for the protected area network. While we attempted to balance computational demands with a sufficient training dataset ($n = 50$ generic parks), we acknowledge that the predictive model is based on a relatively small number of sites. Paired with the number of predictor variables ($n = 8$), we consider that this may limit the generalizability of our model due to potential overfitting and suggest that users keep these potential limits in mind when interpreting their own results.

## Candidate park prioritization

We used the top model according to our model selection to help prioritize a set of real, candidate protected areas across Ontario, Canada ($n = 828$). These included a combination of Ontario Living Legacy sites (OLL; $n = 73$) and Areas of Natural and Scientific Interest (ANSI; $n = 756$). Both sets of sites are natural areas that are not currently regulated as parks, but with features that could lead them to be considered as candidate parks in the future. All of the sites have been identified as important using criteria other than connectivity. To reduce the number of candidate sites, we only included sites with an area equal to or larger than the size of a raster pixel (9 ha), we excluded portions of sites already possessing some degree of protection (i.e., portions overlapping current CPCAD sites), and restricted ANSI's to only provincially significant sites. Following the same methods as with the generic parks above, we calculated values for each of the 8 predictor variables for the set of 828 candidate parks and used this as new input data for the predictive model. The resulting model output was the expected log(ΔMPER) for each of the candidate parks given the values of the 8 predictors and the previously determined model parameter estimates. We then ranked the set of candidate parks based on their expected improvement

to connectivity of the Ontario protected area network. Values of log(ΔMPER) closer to zero translate to higher values of raw ΔMPER, indicating that a given park is expected to have a greater effect on improving network connectivity. We calculated raw ΔMPER values for the set of candidate sites using the correction factor outlined by Sprugel [34] and Baskerville [35] to account for any biases due to the log-transformation used for modelling. To further examine differences between top- and bottom-ranked candidate parks, we overlayed parks in the top 95th percentile ($n=42$) and bottom 5th percentile ($n=42$) onto the current density map of Ontario produced using the current protected areas network [24]. Current density is a mappable output of circuit-theory connectivity analyses that represents the probability of animal movement through a given pixel on the landscape. Pixels with higher current density values have a higher probability of movement through them, and therefore, are more critical for connectivity. We calculated the mean current density within both the top- and bottom-ranked parks and compared these values using a one-sided t-test.

## Results

All 50 generic parks that were added resulted in a decrease in MPER, indicating that they all improved connectivity of the network, however there was variation among the different parks (mean MPER = 243.4966 ohms; range = 243.4941– 243.4967 ohms). We note that the observed changes in MPER were small, but this is not surprising given the addition of small generic parks (area = 6.25km² for all generic parks). Addition of larger parks would be expected to lead to larger reductions in MPER.

The top model according to $AIC_c$ model selection was the global model at the 2 km scale and no other models were within 2 $\Delta AIC_c$ of the top model (S3 Table; [36]). The relationship between most of the predictor variables and log(ΔMPER) was negative, while the effect of proportion of anthropogenic development surrounding generic parks was positive (Table 2). The effects of distance to nearest park, proportion of water surrounding parks and proportion of water within parks were either non-significant, had standard errors overlapping zero, or both (Table 2), which raises uncertainty about these estimates. The estimates from the top model suggest that protected areas with low edge-to-area ratios, are closer to the centre of the landscape and to a sentinel node, have a low degree of human development within their boundaries, but a higher degree of development around them should produce the largest drop in MPER and therefore greatest improvement to connectivity of the network.

The mean expected ΔMPER for the candidate sites was $-1.22 \times 10^{-5}$ with a range = $-3.72 \times 10^{-11} - -3.66 \times 10^{-4}$. Of the 828 candidates evaluated, 42 sites fell within the top 95th percentile in terms of their expected ΔMPER. Many of the top ranked sites were found within the Ontario Shield ecozone, with some also occurring at the edges of the Hudson Bay Lowlands (Fig 2A) and in the Mixedwood Plains (Fig 2B). There were differences in the spatial distributions of top-ranked (top 5%) and bottom-ranked (bottom 5%) parks with top-ranked parks being located across the province (Fig 3; left panel), while most of the bottom-ranked parks were found in the south and a smaller number in the north (Fig 3 right panel). We found that the top-ranked parks had a higher mean current density than the bottom ranked parks ($mean_{top} = 0.63$, $mean_{bottom} = 0.27$, $t = 2.80$, $p < 0.01$; Fig 4), which provides further support for the importance of these sites for connectivity.

## Discussion

We present methods to integrate connectivity into the protected areas planning process by building on the recently developed sentinel node method for evaluating connectivity of protected area networks. Specifically, we developed a predictive model that can be used to prioritize candidate protected areas based on their expected benefit to improving network connectivity as measured by the mean pairwise effective resistance. Our results indicate that protected areas with low edge-to-area ratios, are closer to the centre of the landscape and to a sentinel node, have a low degree of human impact internally, but are adjacent to developed landscapes should have the largest benefit to connectivity of the protected area network. We developed our model using general park characteristics and landscape context variables, and thus we consider it should be broadly applicable across different landscapes and protected area networks.

**Table 2. Model summary for the top regression model for predicting optimal park placement to improve network connectivity. The top model was a global model with all 8 predictor variables at a scale of 2 km for landscape context variables. Significant variables are shown in bold type.**

| Variable | Coefficient | Std. Error | p value |
|---|---|---|---|
| **edge-area-ratio** | **−0.23** | **0.13** | **0.091** |
| **distance to centre** | **−0.005** | **0.002** | **0.003** |
| distance to nearest park | 0.02 | 0.03 | 0.569 |
| **distance to nearest node** | **−0.04** | **0.009** | **<0.001** |
| **proportion of anthropogenic – 2 km buffer** | **4.57** | **1.77** | **0.013** |
| **proportion of anthropogenic – within park** | **−7.26** | **1.78** | **<0.001** |
| proportion of water – 2 km buffer | −3.26 | 2.46 | 0.191 |
| proportion of water – within park | 0.72 | 3.14 | 0.819 |

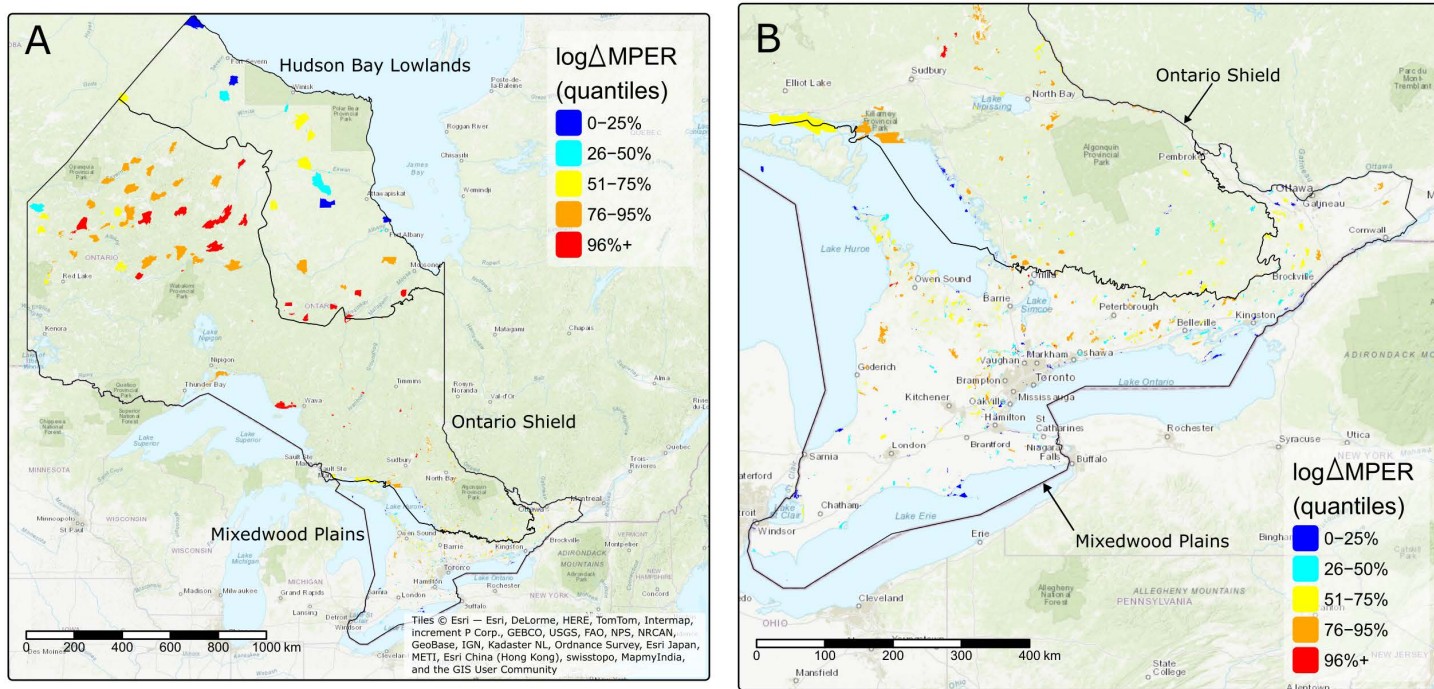

**Fig 2. Candidate protected areas (*n=828*) ranked according to their expected improvement to overall network connectivity.** Values of log(ΔMPER) represent expected changes in overall network connectivity from the addition of a given candidate park to the network where smaller negative values indicate a greater increase in connectivity. Ranked candidate sites are shown A) across the province of Ontario and B) zoomed into southern Ontario for better visualization. Here we group values of log(ΔMPER) into quantile bins where values in the 0–25% range (blue) represent the lowest values and values in the 95th percentile (red) represent the highest values. Thus, candidate parks falling within the 95th percentile should maximize increases in network connectivity. Solid black lines show the boundaries of ecozones in Ontario, Canada. Shapefiles acquired from Ontario Geohub and Statistics Canada and licensed under the Open Government License–Ontario and Open Government Licence–Canada, respectively. Basemaps were obtained using the package tmaptools in the program R [37].

The edge-to-area ratio (EAR) provides a measure of park shape, and our model suggests that generic parks with a lower EAR, lead to greater improvements in network connectivity. While non-significant under a 95% threshold (p<0.05), EAR would be significant under a 90% threshold, which lends support to the importance of this variable for protected area

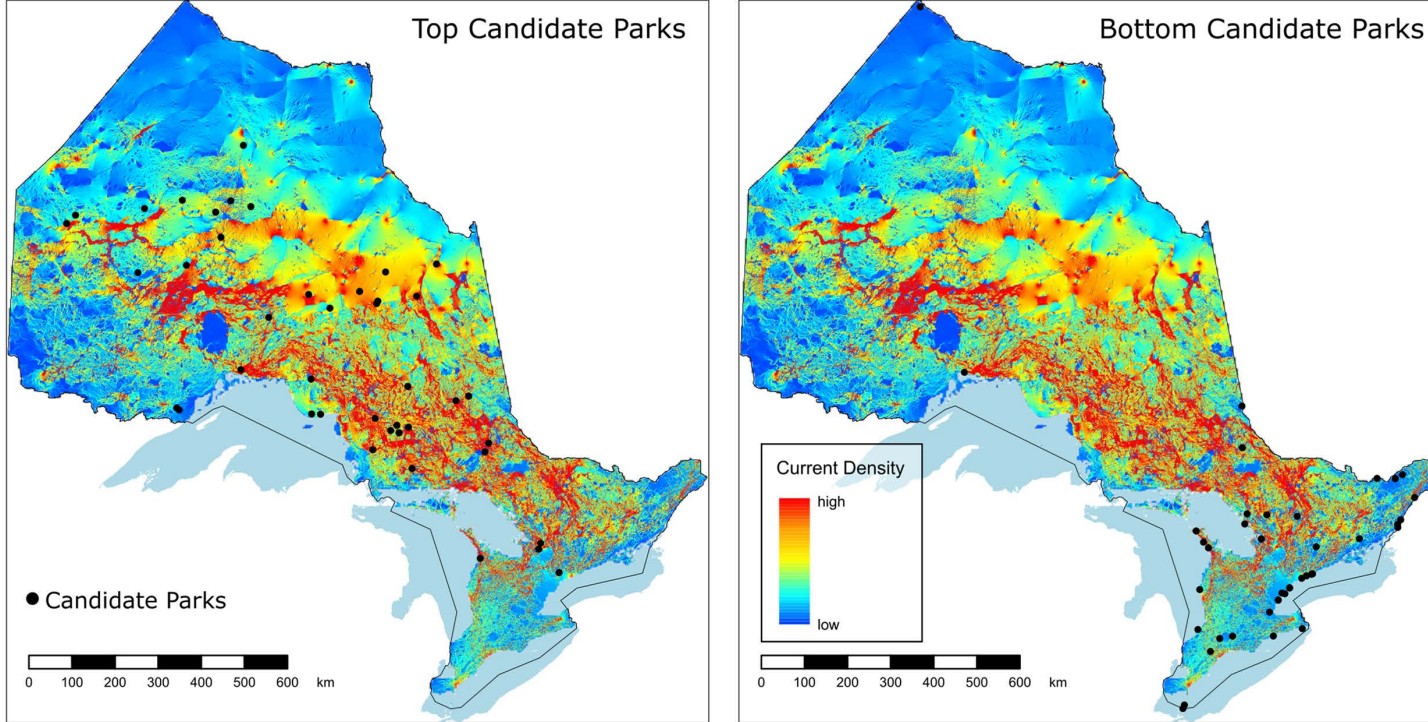

**Fig 3. Current density maps displaying park-to-park connectivity for the protected areas network in Ontario.** Maps show the locations of (left panel) top 5% ($n = 42$) of 828 candidate parks according to predicted benefit to network connectivity and (right panel) bottom 5% ($n = 42$) of candidate parks according to predicted benefit to network connectivity. Current density, measured in amperes, represents the probability of animal movement within a given pixel across the landscape. Higher values of current density represent a higher likelihood of animal movement through a given area on the landscape. Black dots show the locations of candidate parks. Shapefile acquired from Statistics Canada and licensed under the Open Government Licence–Canada.

connectivity. Specifically, this means that protected areas where the difference between the edge length (i.e., perimeter) and the area of the park is minimized, are expected to have a greater benefit for connectivity. This is why the square generic parks generally resulted in lower MPER than the long, narrow rectangular parks. Given these results, we would also expect that a single, large, protected area should be more beneficial to connectivity than a cluster of smaller parks even if they cover the same amount of area since the edge length will be greater for a cluster of small parks. This finding lends some support to the debate of creating 'Single Large OR Several Small' reserves when designing protected area networks [i.e., SLOSS principle; 38] suggesting larger reserves may be best for connectivity, however more recent research suggests the use of 'Single Large AND Several Small' reserves [SLASS; 39]. In fact, Wiersma and Urban [40] suggested that ensuring effective ecological representation in protected area networks may require a combination of large and small reserves when patterns of diversity vary across a landscape. A similar strategy is likely also suitable for designing 'well-connected' protected area networks. For example, a single large, protected area may be most appropriate in intact, natural landscapes, while several smaller protected areas may be more effective at connecting heterogeneous landscapes (e.g., stepping stone patches). Future research could explore how the most effective size of protected area for maintaining connectivity varies with landscape heterogeneity. The relationship between protected area connectivity and species-specific movements could be explored using data from existing animal movement databases (e.g., Movebank).

The negative effects of distance to centre and distance to nearest node were expected given known properties of this type of park-to-park connectivity analysis. Current density is a second output of circuit theory connectivity analyses and

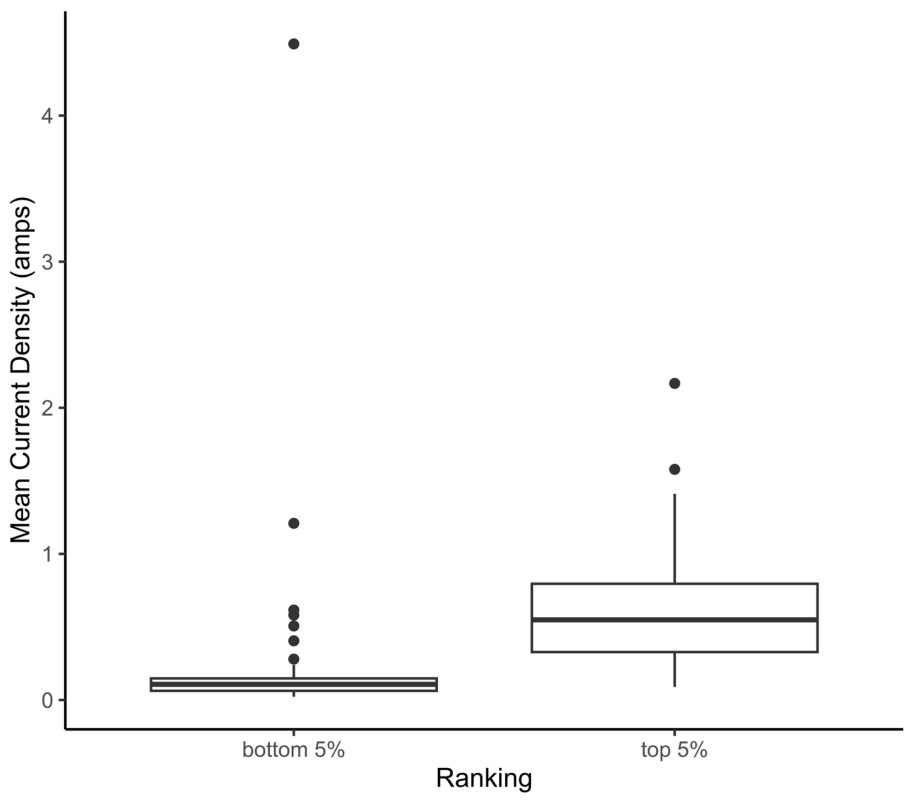

**Fig 4. Differences in mean current density (amps) within top 5% candidate parks (_n_=42) and bottom 5% candidate parks (_n_=42).** Boxplots show the median (thick horizontal line) and central 50% (boxes) of current density values.

measures the probability of animal movement across the landscape. The higher the current density for a given area, the higher the likelihood of an animal moving through that area. However, one artifact of park-to-park connectivity analyses is that because the nodes are contained within the study area, high current density tends to pool in the centre of the study area. As a result of this, we would expect that parks added further from the centre would have less of an impact on improving connectivity than parks added closer to the centre where current density is generally higher. In addition, high current density also tends to form around the nodes themselves [28]. Thus, we would also expect the negative effect of distance to the nearest sentinel nodes where parks added closer to a sentinel node should lead to a greater reduction in MPER. We included these variables in our model to help account for potential biases that may exist and advise that practitioners using our model for planning of their own protected area network should consider these biases when interpreting model results.

Taken in tandem, the negative effect of human development within parks and positive effect of development around parks can again be explained by known properties of circuit theoretic connectivity models. It has been shown that the close proximity of low cost (i.e., natural cover) and high cost (e.g., roads, cities, lakes) landcover features leads to a funneling of high current density and therefore higher probability of animal movement [41], often referred to as 'pinch-points'. This is because movement through these low-cost areas is constricted by the adjacent high cost landcover. In the case of new protected areas, the results of our model suggest that protecting natural areas that are adjacent to areas with a higher degree of development should have a larger benefit to connectivity of the overall network. There are fewer opportunities for movement within these heterogeneous landscapes and so natural areas here are more critical for movement

[42]. In contrast, addition of a protected area to a large intact landscape, may not be expected to be as beneficial given the similarities in movement potential inside and outside of the protected area. We note that these intact landscapes are still critically important to protect to maintain connectivity and to protect the innumerable benefits of intact landscapes for species restricted in intact habitat, biodiversity, and human well-being [43,44]. We would have expected water to have a similar effect on the landscape as anthropogenic development given the high cost to movement, however, both water variables were found to be non-significant. This result is likely due to the high variability of water bodies on the landscape leading to varied effects on connectivity, which is apparent from the large confidence intervals.

We tested our model as a predictive tool by providing new input data to evaluate a set of 828 real, candidate protected area sites across Ontario that were previously identified using criteria other than connectivity value. The resulting model output represents the predicted benefit to network connectivity of adding a given site to the current protected area network. In other words, how much would we expect a given candidate site to reduce the mean pairwise effective resistance of the network. Protection of the highest ranked sites should therefore provide the greatest increase in connectivity of the Ontario protected area network. While we used previously identified sites, going forward, our model could be used to evaluate any parcel of land across a given landscape with defined boundaries. With an understanding of the features that best predict park placement according to our model, new sites to evaluate could be selected by choosing sites with a low edge-to-area ratio, considering distance to the closest sentinel node and to the centre of the landscape, and looking at maps of landcover. Additionally, future sites could also be identified by looking at a map of current density in which pixel values represent the probability of animal movement through a given pixel. Areas with high current density values are likely important sites to consider for future protection. Indeed, we found that the top sites predicted by our model were found in areas with higher mean current density than those ranked at the bottom, which provides validation for our results.

Incorporating connectivity into conservation planning requires access to technical documentation, guidelines on best practices, and simple to use tools [45]. Our method adds to the collection of existing systematic conservation planning methods and tools that can integrate connectivity into spatial prioritization and protected areas planning [45,46]. Conservation planning software, such as Marxan Connect [45] and Zonation [47], give users the option of including connectivity as a feature in the prioritization process and aim to balance conservation benefits with design costs. Here, we provide a simple operational tool to translate connectivity metrics into planning-relevant rankings, which can be used to prioritize sites based on their expected benefit to network connectivity. While these other tools allow for the optimization of multiple conservation features, our predictive model allows users to generate a fast and simple calculation of a connectivity ranking, without the often computationally demanding step of running a connectivity model.

Integrating functional connectivity into reserve design is considered ideal given the species-specific nature of connectivity (e.g., 46), however, the data needed to measure functional connectivity is rarely available and especially at the scale required [48]. As such, our method was developed using a multispecies connectivity approach with the intent that our model would be broadly applicable to the conservation needs of a range of terrestrial, non-volant species [32]. The benefits of a multispecies, naturalness approach include a reduced financial burden, lower time commitment, and such models can also facilitate simpler decision-making amongst multiple stakeholder groups [49]. Compared to other existing structural connectivity metrics (e.g., ProNet [20]; Integrated Index of Connectivity [50]), which estimate how well-connected a protected area network is based on area and distance (Euclidian or resistance) between sites, our model output is used to rank potential sites based on their expected connectivity benefit. The sentinel node method outlined by O'Brien et al. [24] and the predictive model outlined here provide a toolbox for conservation practitioners to evaluate and monitor connectivity of their protected area systems and identify priority sites to help improve upon connectivity of the network. The choice of which connectivity metrics and methods to use will likely depend on the scale and objectives of the conservation strategy and practitioners should make use of existing documentation to help select the most appropriate method for their needs [e.g., 10,51]. However, we consider that our predictive model should be useful across a range of scales and objectives from prioritizing land parcels by local land trust organizations to creating a well-connected park system (including

Indigenous, provincial/state, and national parks). Bridging the gap between research and decision-makers is critical for advancing conservation in practice. As such, we have directly shared this tool and associated guidelines with relevant stakeholders to ensure it can be incorporated by those involved in protected area and land-use planning decisions.

We acknowledge that our method does have limitations that should be considered by potential users. First, we only considered connectivity under current conditions and did not account for future climatic conditions [e.g., 52] or future development. Therefore, areas predicted to be important by our model under current conditions only provide a snapshot of current connectivity and may not hold under future conditions such as further fragmentation (road development, etc.) and a different landscape vegetatively due to climatic changes. We suggest re-evaluating connectivity and updating models every 10 years in alignment with provincial timelines for reporting on the state of its protected areas. Second, the model was built using 300-m resolution data, which we felt was an appropriate scale for the objectives and spatial extent of our study. Thus, our model should be applicable to other landscapes at the same scale. Those interested in using a higher resolution of data to meet their needs should consider re-evaluating models to determine how effects may vary with scale, especially landscape context variables. We determined that a 2-km buffer was the most appropriate scale to evaluate the effects of landscape context surrounding protected areas, however, this is likely to vary with the resolution of the underlying cost surface. Finally, we did not consider land tenure in our model, and so the resulting output does not distinguish between private and public land. This is less of a limitation and more of a reminder for users to consider prior to modelling based on their objectives and jurisdiction (e.g., local land trust organization versus regional government). Additionally, it is important to consider multiple criteria in the planning process and we suggest that the output of our model could easily be integrated with other layers of interest such as land costs [27], climate change mitigation [53], or biodiversity [54].

## Conclusion

Target 3 of the Kunming-Montreal Agreement commits countries to protecting 30% of terrestrial lands in systems of well-connected protected areas. As nations pushes to meet the ambitious 30 × 30 target, it will be critical not only to increase area-based protection, but also to ensure these protected areas are functionally and structurally connected. We extend the sentinel node framework by developing a predictive model for prioritization. Our model can be used to assess candidate protected areas according to their expected benefit to network connectivity based on a suite of park and landscape characteristics. We consider that this will be an especially useful tool for future protected areas development by helping to guide decision-making by diverse land managers and stakeholders and allowing explicit incorporation of connectivity into the planning process in a way that can complement existing priorities. Evaluation of potential protected areas according to their connectivity value should ensure that new protected areas help to improve connectivity of the overall network and thus help make meaningful contributions towards the 30 × 30 target.

## Supporting information

**S1 Table. List of cost values used to classify the cost surface and types of landscape features assigned each cost rank.** See Pither et al. [31] for a detailed description of landscape feature classifications and data layers used. (DOCX)

**S2 Table. Descriptions of the 10 candidate linear regression models evaluated to determine the best predictors of protected area placement for improving connectivity.** Buffer size indicates the size of buffer placed around generic parks when calculating landscape context variables. A range of buffer sizes were tested to determine the scale of effect of landscape context variables. Model variable codes represent: EAR = edge-to-area ratio; Dist_Cent = distance to center; Dist_Park = distance to nearest park; Dist_Node = distance to nearest sentinel node; PropAnth_Park = proportion of anthropogenic development within parks; PropWater_Park = proportion of water within parks; PropAnth_# = proportion of

anthropogenic development within a buffer surrounding parks (one of 2, 10, 25, or 50 km); and PropWater_#=proportion of water within a buffer surrounding parks (one of 2, 10, 25, or 50 km).
(DOCX)

**S3 Table. Summary of AIC$_c$ model selection results.** Model number corresponds to those used in S2 Table
(DOCX)

## Acknowledgments

Thank you to additional members of the Ontario Parks connectivity working group Karen Hartley, Louis Chora and Amanda Schroeder for advice. We also thank Richard Pither, Angela Brennan, and Kristen Hirsh-Pearson for collaboration on related work.

## Author contributions

**Conceptualization:** Paul O'Brien, Natasha Carr, Jeff Bowman.

**Data curation:** Paul O'Brien.

**Formal analysis:** Paul O'Brien.

**Investigation:** Paul O'Brien.

**Methodology:** Paul O'Brien, Jeff Bowman.

**Supervision:** Jeff Bowman.

**Visualization:** Paul O'Brien.

**Writing – original draft:** Paul O'Brien, Natasha Carr, Jeff Bowman.

**Writing – review & editing:** Paul O'Brien, Natasha Carr, Jeff Bowman.

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
