## [Decision Letter · Decision Letter 0]

22 Aug 2025

Dear Dr.  O'Brien,

Thank you for submitting your manuscript to PLOS ONE. After careful consideration, we feel that it has merit but does not fully meet PLOS ONE’s publication criteria as it currently stands. Therefore, we invite you to submit a revised version of the manuscript that addresses the points raised during the review process.

We look forward to receiving your revised manuscript.

Kind regards,

Daniel de Paiva Silva, Ph.D.

Academic Editor

PLOS ONE

Journal Requirements:

“Funding for this research was provided by the Ontario Ministry of the Environment, Conservation, and Parks and the Ontario Ministry of Natural Resources.”

4. We note that Figure 1, 2 and 3 in your submission contain map images which may be copyrighted. All PLOS content is published under the Creative Commons Attribution License (CC BY 4.0), which means that the manuscript, images, and Supporting Information files will be freely available online, and any third party is permitted to access, download, copy, distribute, and use these materials in any way, even commercially, with proper attribution. For these reasons, we cannot publish previously copyrighted maps or satellite images created using proprietary data, such as Google software (Google Maps, Street View, and Earth). For more information, see our copyright guidelines: http://journals.plos.org/plosone/s/licenses-and-copyright.

1. You may seek permission from the original copyright holder of Figure 1, 2 and 3                                                               to publish the content specifically under the CC BY 4.0 license.

Additional Editor Comments:

Dear Dr. O'Brien,

After this review round, both reviewers believe your manuscript is almost ready for acceptance. Therefore, please aply the suggestions made by the reviewers and resubmit your manuscript and I am sure the text will be accepted for publication in PLoS One.

Sincerely,

Daniel Silva

Reviewer's Responses to Questions

**Comments to the Author**

1. Is the manuscript technically sound, and do the data support the conclusions?

Reviewer #1: Yes

Reviewer #2: Yes

2. Has the statistical analysis been performed appropriately and rigorously?

Reviewer #1: Yes

Reviewer #2: Yes

3. Have the authors made all data underlying the findings in their manuscript fully available?

Reviewer #1: Yes

Reviewer #2: Yes

4. Is the manuscript presented in an intelligible fashion and written in standard English?

Reviewer #1: Yes

Reviewer #2: Yes

Reviewer #1: I enjoyed reading the manuscript by O'Brien and collaborators. They studied an alternative method to predict protected areas (PAs) network connectivity, measured connectivity using simulated PAs, and provided PA characteristics that benefit connectivity. The density values maps were an excellent output of the tool presented and are very suitable for application in decision-making on conservation planning. The manuscript is well-written and has a robust general design.

After addressing a few issues, I found this article suitable for publication in Plos One.

One of my concerns is the lack of a practical perspective on how this method (and tool) will be made available to stakeholders who are central to protected area planning. While I understand that this was not the primary goal of the study, providing insights into how the tool will be delivered to key stakeholders could help bridge the gap between academic ideas and practical implementation. Without this, the method risks being another valuable yet underutilized concept published in the literature.

Additionally, some information requires a detailed explanation instead of being limited to a single sentence or citation. For instance, "well-connected" and Canadian protected area networks. The first terminology appears multiple times in the text, however, what is its meaning? It would be helpful to specify the criteria or requirements for connectivity in this context. For the second, it would be beneficial for international readers to include a brief explanation of how PA establishment works in Canada, as this process varies significantly between countries. Further, the five cost ranks were not well explained. 0.1 is the value within the PA and 1000 is the value on a very anthropogenic landscape, but the intermediate values remain unclear. Finally, a schematic figure summarizing the methods could be a valuable addition, helping readers better understand the sequence of analyses performed.

In the next few lines, I detail smaller issues:

Abstract - concise and well-written. The only lack is to provide a practical view of how this tool will be available to the decision-making actors.

L.83 - "(in fact, likely the opposite)" - Require an explanation.

L.106 / 119- The terminology "well-connected" needs to be explained.

L.124 -Please, provide the reader with the difference between generic parks and PA networks. Were generic parks the simulated ones? The sentence is not clear.

L. 128 - Why did you prefer to use random PAs within the study site than to choose characteristics that are important (size, edge, distance to other PA, etc) to PA effectiveness?

L. 128-133- The PAs establishment is different around the world. For comparison reasons, it is essential to provide details on how this works in Canada. In addition, in L.212-213, the broad readership does not know about it.

L. 137 - Provide information on the 5 cost ranks. Include it in the legend of Figure 1.

L. 159 - Why did you choose to work with small-sized (6.25 km²) PAs?

L. 180 - Which ones?

L. 248-249 - Interesting result. A discussion on this is needed.

L. 295-298 - Repeated information, is not necessary.

L. 306-308 - I agree. However, to make the bridge between academics and conservation in practice, how to deliver this approach in a practical way?

L. 317-327 - To this discussion, movement data (Movebank database) on known species could be simulated, and this may be the next step of this study.

L. 332-337 - Is there any more bias linked to the nodes? This info may fit better in the Method Section.

L. 393- 410 - Very nice paragraph.

Figure 2 needs quality improvements.

Table S1 fits better in the main text, very important information.

Reviewer #2: General Evaluation

The manuscript reports original research on integrating connectivity measures into conservation planning. The study presents relevant results that may inspire replication in other contexts. The proposed strategy for addressing functional connectivity can be objectively implemented into conservation/restoration targets, and thus constitutes a valuable contribution to conservation science.

The manuscript has significant merit and should be considered for publication in PLOS One after the authors address the methodological clarifications, improve figure quality, and strengthen the interpretation of results. Below I provide some specific recommendations:

Materials and Methods

Line 129: Provide a clear rationale for the selection of 20 sentinel nodes. Explain how this choice could be adapted in landscapes with different numbers of nodes or spatial extents (e.g., 2,500 nodes or 50 nodes). Clarify whether micro-fragments are included.

Line 137: Explicitly classify all cost values. Which correspond to natural areas (e.g., 0.1), and what about the others?

Lines 156–157: Specify whether the 50 new generic parks represent intact natural areas or partially degraded areas. Detail the criteria for distance within the randomization process.

Line 156: Remove the term “randomly”, since the parks were selected according to a 50 km distance criterion and are evenly distributed. Justify why 50 parks were chosen.

Lines 163–165: Specify the basis for the variety of possible additions of protected areas (e.g., vegetation type).

Lines 165–168: Revise this unclear sentence. If generic parks reduce costs from 1 to 0.1, but natural areas were already 0.1, clarification is needed.

Figures and Tables

Figure 1: Improve resolution and add labels directly in the image. Present one map with natural areas, sentinel nodes, and generic parks overlaid, and another with cost analysis results.

Figure 2: Replace with a higher-resolution map.

Figure 3: Make the map more self-explanatory by replacing “A”/“B” with “top candidate parks” / “bottom candidate parks” and labeling black dots as “candidate parks”.

Table 1: Report that the edge-area-ratio yielded p = 0.09 under a 95% threshold (non-significant), but would be significant under a 90% threshold. This should be mentioned and its implications discussed.

Discussion

Lines 303–304: Avoid using sentinel node proximity as a conservation parameter, as sentinel nodes are analytical constructs.

Lines 317–318: Specify which side of the SLOSS debate the results support (single large vs. several small).

Lines 317–327: Specify which connectivity model is most suitable for the study region.

Lines 354–356: Specify that, while natural areas adjacent to developed areas may contribute more to increasing overall connectivity, intact areas remain essential under other conservation criteria (e.g., species restricted to intact habitats).

Lines 395–397: Revise the apparent contradiction regarding temporal applicability. Specify the temporal scale at which the method can project future scenarios (medium- or long-term), and what adjustments would be needed to account for future conditions.

Conclusion

Emphasize how this approach differs from existing methods (e.g., Integral Index of Connectivity in Conefor, or direct resistance values).

Highlight the novelty of prioritizing areas based on their potential to increase overall connectivity when added to the conservation network.

.

Reviewer #1: **Yes:** Bertassoni, A.Bertassoni, A.Bertassoni, A.Bertassoni, A.

Reviewer #2: No

While revising your submission, please upload your figure files to the Preflight Analysis and Conversion Engine (PACE) digital diagnostic tool, https://pacev2.apexcovantage.com/. PACE helps ensure that figures meet PLOS requirements. To use PACE, you must first register as a user. Registration is free. Then, login and navigate to the UPLOAD tab, where you will find detailed instructions on how to use the tool. If you encounter any issues or have any questions when using PACE, please email PLOS at . PACE helps ensure that figures meet PLOS requirements. To use PACE, you must first register as a user. Registration is free. Then, login and navigate to the UPLOAD tab, where you will find detailed instructions on how to use the tool. If you encounter any issues or have any questions when using PACE, please email PLOS at . PACE helps ensure that figures meet PLOS requirements. To use PACE, you must first register as a user. Registration is free. Then, login and navigate to the UPLOAD tab, where you will find detailed instructions on how to use the tool. If you encounter any issues or have any questions when using PACE, please email PLOS at . PACE helps ensure that figures meet PLOS requirements. To use PACE, you must first register as a user. Registration is free. Then, login and navigate to the UPLOAD tab, where you will find detailed instructions on how to use the tool. If you encounter any issues or have any questions when using PACE, please email PLOS at figures@plos.org. Please note that Supporting Information files do not need this step.. Please note that Supporting Information files do not need this step.

---

## [Author Response · Author response to Decision Letter 1]

21 Nov 2025

Journal Requirements:

Response 1: We have revised the manuscript to meet style requirements.

“Funding for this research was provided by the Ontario Ministry of the Environment, Conservation, and Parks and the Ontario Ministry of Natural Resources.”

Response 2: We have included the adjusted financial statement in our cover letter as suggested.

Response 3: Our data is stored on an online repository as indicated in our data availability statement and will be publicly available upon acceptance of the manuscript.

4. We note that Figure 1, 2 and 3 in your submission contain map images which may be copyrighted. All PLOS content is published under the Creative Commons Attribution License (CC BY 4.0), which means that the manuscript, images, and Supporting Information files will be freely available online, and any third party is permitted to access, download, copy, distribute, and use these materials in any way, even commercially, with proper attribution. For these reasons, we cannot publish previously copyrighted maps or satellite images created using proprietary data, such as Google software (Google Maps, Street View, and Earth). For more information, see our copyright guidelines: http://journals.plos.org/plosone/s/licenses-and-copyright.

Response 4: We have included the necessary attribution statements for these figures within the figure captions and directly within the map for Figure 2.

Response 5: We have moved the Supporting Information section to the end as suggested.

Response 6: No citations were recommended.

Response 7: We have reviewed our reference list, and everything is complete and correct.

Additional Editor Comments:

Dear Dr. O'Brien,

After this review round, both reviewers believe your manuscript is almost ready for acceptance. Therefore, please apply the suggestions made by the reviewers and resubmit your manuscript and I am sure the text will be accepted for publication in PLoS One.

Sincerely,

Daniel Silva

Reviewer #1:

I enjoyed reading the manuscript by O'Brien and collaborators. They studied an alternative method to predict protected areas (PAs) network connectivity, measured connectivity using simulated PAs, and provided PA characteristics that benefit connectivity. The density values maps were an excellent output of the tool presented and are very suitable for application in decision-making on conservation planning. The manuscript is well-written and has a robust general design.

Response 8: We are glad to hear the reviewer enjoyed our manuscript and appreciate the provided suggestions. We have done our best to incorporate all suggestions, which we think has greatly improved our manuscript.

After addressing a few issues, I found this article suitable for publication in Plos One.

One of my concerns is the lack of a practical perspective on how this method (and tool) will be made available to stakeholders who are central to protected area planning. While I understand that this was not the primary goal of the study, providing insights into how the tool will be delivered to key stakeholders could help bridge the gap between academic ideas and practical implementation. Without this, the method risks being another valuable yet underutilized concept published in the literature.

Additionally, some information requires a detailed explanation instead of being limited to a single sentence or citation. For instance, "well-connected" and Canadian protected area networks. The first terminology appears multiple times in the text, however, what is its meaning? It would be helpful to specify the criteria or requirements for connectivity in this context. For the second, it would be beneficial for international readers to include a brief explanation of how PA establishment works in Canada, as this process varies significantly between countries. Further, the five cost ranks were not well explained. 0.1 is the value within the PA and 1000 is the value on a very anthropogenic landscape, but the intermediate values remain unclear. Finally, a schematic figure summarizing the methods could be a valuable addition, helping readers better understand the sequence of analyses performed.

Response 9: Thank you for bringing these issues to our attention. We have done our best to address these below in the reviewer’s specific comments.

In the next few lines, I detail smaller issues:

Abstract - concise and well-written. The only lack is to provide a practical view of how this tool will be available to the decision-making actors.

Response 10: Thank you. We agree that a practical view of how this tool will be made available is important. We consider that we do not have space to add this to the abstract, but we have added some information on this in the discussion (L428-431).

L.83 - "(in fact, likely the opposite)" - Require an explanation.

Response 11: We have added some explanation here (L82-83).

L.106 / 119- The terminology "well-connected" needs to be explained.

Response 12: We have provided a definition at L88-91.

L.124 -Please, provide the reader with the difference between generic parks and PA networks. Were generic parks the simulated ones? The sentence is not clear.

Response 13: We have added some clarification here (L126).

L. 128 - Why did you prefer to use random PAs within the study site than to choose characteristics that are important (size, edge, distance to other PA, etc) to PA effectiveness?

Response 14: As outlined in previous research (O’Brien et al. 2023), we preferred a random selection procedure of sentinel node parks to ensure a relatively even distribution of nodes across the landscape rather than selecting parks based on other criteria (e.g., size), which may lead to a bias in node location and clustering of nodes. We have added a sentence to clarify this point (L132-135).

L. 128-133- The PAs establishment is different around the world. For comparison reasons, it is essential to provide details on how this works in Canada. In addition, in L.212-213, the broad readership does not know about it.

Response 15: Thank you for the suggestion. We think that it is not essential here to understand PA establishment in Canada, given that many PAs can be national, provincial/territorial, and private, and so there is not a single “Canadian” process for PA establishment. However, we have now provided a short description of the Canadian Protected Areas Database (CPCAD) to help clarify to the broad readership what this includes (L140-142), to help understand the Canadian context.

As for L212-213, we consider the exact details of these sites is not critical, but most important as we outline in the following lines (L235-237) that they represent “natural areas that are not currently protected but have been identified using characteristics making them suitable for protection”. In addition, we have now provided a link which provides more detail for the Areas of Natural and Scientific Interest (ANSIs; L234). Hopefully this provides more clarity to those readers who may be interested.

L. 137 - Provide information on the 5 cost ranks. Include it in the legend of Figure 1.

Response 16: We have now included a table in the supplemental material (S1 Table) that provides more details on the 5 cost ranks.

L. 159 - Why did you choose to work with small-sized (6.25 km²) PAs?

Response 17: As described in L176-177, we selected this size of park to match the average size of PAs in this region, which tend to be small.

L. 180 - Which ones?

Response 18: Assuming the reviewer is asking which generic parks were added, we added each to the landscape independently of one another to determine how the addition of each influenced connectivity of the PA network. Or perhaps the reviewer is wondering if the generic parks are the simulated or real parks? If so, we have added text to clarify that here we are referring to the simulated parks (L197).

L. 248-249 - Interesting result. A discussion on this is needed.

Response 19: Indeed, we agree this warrants some discussion. We have added some lines interpreting these results to the discussion (L383-387).

L. 295-298 - Repeated information, is not necessary.

Response 20: We have removed this as suggested.

L. 306-308 - I agree. However, to make the bridge between academics and conservation in practice, how to deliver this approach in a practical way?

Response 21: This is a great point. We have added some discussion of this at L428-431. We also note that we have already shared this tool directly with stakeholders through a series of workshops, which we highlight now (L429-431). The code and data required to run the predictive model described in this manuscript is also available online at https://figshare.com/s/17e67a7fafa68b5af9ec, which we provide the ‘Data Availability’ section.

L. 317-327 - To this discussion, movement data (Movebank database) on known species could be simulated, and this may be the next step of this study.

Response 22: Indeed, the effectiveness of different sized protected areas in terms of connectivity is likely to vary with species. We have added some text to discuss how this species-specific relationship could be explored in the future (L353-354).

L. 332-337 - Is there any more bias linked to the nodes? This info may fit better in the Method Section.

Response 23: Given this information is discussing results of the modelling, we consider that it makes sense to keep this info in the discussion rather than move to the methods.

L. 393- 410 - Very nice paragraph.

Response 24: Thank you.

Figure 2 needs quality improvements.

Response 25: We have done this.

Table S1 fits better in the main text, very important information.

Response 26: We have moved this table to the main text as suggested. This is now Table 1 in the revised manuscript

Reviewer #2:

General Evaluation

The manuscript reports original research on integrating connectivity measures into conservation planning. The study presents relevant results that may inspire replication in other contexts. The proposed strategy for addressing functional connectivity can be objectively implemented into conservation/restoration targets, and thus constitutes a valuable contribution to conservation science.

The manuscript has significant merit and should be considered for publication in PLOS One after the authors address the methodological clarifications, improve figure quality, and strengthen the interpretation of results. Below I provide some specific recommendations:

Response 26: Thank you for your kind words. We have done our best to address your recommendations, which we think has greatly improved our manuscript.

Materials and Methods

Line 129: Provide a clear rationale for the selection of 20 sentinel nodes. Explain how this choice could be adapted in landscapes with different numbers of nodes or spatial extents (e.g., 2,500 nodes or 50 nodes). Clarify whether micro-fragments are included.

Response 27: We followed Koen et al. (2014), who showed that 15-20 nodes was sufficient for modelling omnidirectional connectivity across a similar region. We have added some explanation of this at L135-138. We suspect that larger landscapes (i.e., with more pixels) may require more nodes, but this is a research question that remains to be answered. Regarding the inclusion of micro-fragments, we considered a protected area for selection as a sentinel node so long as it was larger than or equal to the size of a pixel (i.e., 100m2). We have added a line to clarify this (L131-132).

Line 137: Explicitly classify all cost values. Which correspond to natural areas (e.g., 0.1), and what about the others?

Response 28: Please see Response 16 to Reviewer 1.

Lines 156–157: Specify whether the 50 new generic parks represent intact natural areas or partially degraded areas. Detail the criteria for distance within the randomization process.

Response 29: The 50 generic parks contained a mix of natural and developed areas. The only criteria was that the centroid pixel be classified as natural. We have added some text to clarify this (L171-173). The randomization algorithm used a 50km distance value to ensure selected pixels were at least 50km apart. This was to ensure that there was no overlap between generic parks when calculating landscape variables surrounding parks. We have clarified this at L168-175.

Line 156: Remove the term “randomly”, since the parks were selected according to a 50 km distance criterion and are evenly distributed. Justify why 50 parks were chosen.

Response 30: We have kept the term “randomly” because parks were still selected according to a stratified random procedure from available spatial points. Specifically, our algorithm randomly selected 50 spatial points where a stratum was applied to the selection procedure, ensuring that points were at least 50 km apart. We have added text to clarify this in the manuscript (L169-171). We have also removed the term “evenly”, which has given the impression that generic parks were spaced exactly 50km apart from each other when they were actually at least 50km apart, but potentially more.

50 generic parks were selected because we considered that would provide a sufficient sample size for our models and would not be too computationally demanding. We have added some text on this justification (L166-167).

---

## [Decision Letter · Decision Letter 1]

27 Jan 2026

Dear Dr. O'Brien,

Thank you for submitting your manuscript to PLOS ONE. After careful consideration, we feel that it has merit but does not fully meet PLOS ONE’s publication criteria as it currently stands. Therefore, we invite you to submit a revised version of the manuscript that addresses the points raised during the review process.

We look forward to receiving your revised manuscript.

Kind regards,

Daniel de Paiva Silva, Ph.D.

Academic Editor

PLOS One

Journal Requirements:

Additional Editor Comments:

Dear Dr. O'Brien,

After this new review round, the reviewer believes your work is nearly suitable for publication in PLoS One after minor reviews are implemented. I hope the provided suggestions help you to improve your text. Once you finish implementing them, the work will be ready to be sent out for production.

Sincerely,

Daniel Silva

Reviewers' comments:

Reviewer's Responses to Questions

**Comments to the Author**

Reviewer #2: (No Response)

2. Is the manuscript technically sound, and do the data support the conclusions?

Reviewer #2: Yes

3. Has the statistical analysis been performed appropriately and rigorously?

Reviewer #2: Yes

4. Have the authors made all data underlying the findings in their manuscript fully available?

Reviewer #2: Yes

5. Is the manuscript presented in an intelligible fashion and written in standard English?

Reviewer #2: Yes

Reviewer #2: The manuscript PONE-D-24-48938R1 presents important results for implementing connectivity in conservation planning, with strong policy relevance and timely alignment with the Kunming–Montreal GBF and 30 × 30 targets. It also presents a clear methodological framework with strong practical applicability for managers and practitioners. I recommend acceptance of the manuscript in PLOS ONE, following minor revisions to address some writing details. Below I provide several comments that can be addressed with additional clarification in the text:

The manuscript presents an approach that is theoretically well established in the connectivity and conservation planning literature. However, it provides a meaningful methodological increment by integrating techniques such as the use of sentinel nodes, MPER, simulations, and regression to predict connectivity gains from new protected areas. This methodological extension is particularly valuable for practical applications, as it creates a simplified model that managers can use to prioritize candidate areas based on site and landscape attributes. I believe that the methodological novelty is not sufficiently clear in the Introduction, which may confuse readers, given that the underlying techniques and metrics are well known.

To better specify the innovation of the study, I suggest replacing the phrases “we developed a novel predictive model” (lines 454–455) and “we highlight the novelty of our approach” (line 411) with “we extend the sentinel node framework by developing a predictive surrogate model for prioritization” and “we provide an operational tool to translate connectivity metrics into planning-relevant rankings”, respectively.

The cost–benefit relationship of this approach relative to other techniques (e.g., Marxan and Zonation) is not clearly discussed and should be clarified in the Discussion.

The predictive model is based on a relatively small training dataset (n = 50, with 8 predictors), which may limit its generalizability due to potential overfitting. I suggest acknowledging this more explicitly in the Methods.

The title would benefit from emphasizing the practical applicability of the study, for example: “A predictive approach for incorporating connectivity into protected areas planning”.

.

Reviewer #2: No

---

## [Author Response · Author response to Decision Letter 2]

17 Mar 2026

Dear Dr. O'Brien,

After this new review round, the reviewer believes your work is nearly suitable for publication in PLoS One after minor reviews are implemented. I hope the provided suggestions help you to improve your text. Once you finish implementing them, the work will be ready to be sent out for production.

Sincerely,

Daniel Silva

Response 1: We thank the editor for the opportunity to revise our manuscript. We think the provided suggestions have greatly improved the manuscript. We look forward to continuing the process towards publication at PloS One.

Reviewers' comments:

Reviewer #2:

The manuscript PONE-D-24-48938R1 presents important results for implementing connectivity in conservation planning, with strong policy relevance and timely alignment with the Kunming–Montreal GBF and 30 × 30 targets. It also presents a clear methodological framework with strong practical applicability for managers and practitioners. I recommend acceptance of the manuscript in PLOS ONE, following minor revisions to address some writing details. Below I provide several comments that can be addressed with additional clarification in the text:

Response 2: We are delighted to hear that the reviewer sees the importance and practical applicability of our method, and we thank them for their insightful comments on our manuscript. We have done our best to address these revisions below, which we consider has improved our manuscript.

The manuscript presents an approach that is theoretically well established in the connectivity and conservation planning literature. However, it provides a meaningful methodological increment by integrating techniques such as the use of sentinel nodes, MPER, simulations, and regression to predict connectivity gains from new protected areas. This methodological extension is particularly valuable for practical applications, as it creates a simplified model that managers can use to prioritize candidate areas based on site and landscape attributes. I believe that the methodological novelty is not sufficiently clear in the Introduction, which may confuse readers, given that the underlying techniques and metrics are well known.

To better specify the innovation of the study, I suggest replacing the phrases “we developed a novel predictive model” (lines 454–455) and “we highlight the novelty of our approach” (line 411) with “we extend the sentinel node framework by developing a predictive surrogate model for prioritization” and “we provide an operational tool to translate connectivity metrics into planning-relevant rankings”, respectively.

Response 3: Thank you for the suggestion. We have added this new text (L412-414 & L467-468).

The cost–benefit relationship of this approach relative to other techniques (e.g., Marxan and Zonation) is not clearly discussed and should be clarified in the Discussion.

Response 4: We have added some text at lines 406-441.

The predictive model is based on a relatively small training dataset (n = 50, with 8 predictors), which may limit its generalizability due to potential overfitting. I suggest acknowledging this more explicitly in the Methods.

Response 5: We have added more text to the Methods as suggested (L227-232)

The title would benefit from emphasizing the practical applicability of the study, for example: “A predictive approach for incorporating connectivity into protected areas planning”.

Response 6: We have modified the title as suggested.

---

## [Editor Report · Decision Letter 2]

18 Mar 2026

A predictive approach to integrating connectivity into landscape scale protected areas planning

PONE-D-24-48938R2

Dear Dr. O'Brien,

We’re pleased to inform you that your manuscript has been judged scientifically suitable for publication and will be formally accepted for publication once it meets all outstanding technical requirements.

Kind regards,

Daniel de Paiva Silva, Ph.D.

Academic Editor

PLOS One

Additional Editor Comments (optional):

Dear Dr. O' Brien,

I am pleased to accept your manuscript for publication in PLoS One! Congratulations on the hard work you and your co-authors employed on improving this contribution.

Sincerely,

Daniel Silva
---

## [Editor Report · Acceptance letter]

PONE-D-24-48938R2

PLOS One

Dear Dr. O'Brien,

I'm pleased to inform you that your manuscript has been deemed suitable for publication in PLOS One. Congratulations! Your manuscript is now being handed over to our production team.

Kind regards,

on behalf of

Dr. Daniel de Paiva Silva

Academic Editor

PLOS One